# Quantum-Safe Group Key Establishment Protocol from Lattice Trapdoors

**DOI:** 10.3390/s22114148

**Published:** 2022-05-30

**Authors:** Teklay Gebremichael, Mikael Gidlund, Gerhard P. Hancke, Ulf Jennehag

**Affiliations:** 1Department of Information Systems and Technology, Mid Sweden University, 852 30 Sundsvall, Sweden; mikael.gidlund@miun.se; 2Department of Computer Science, City University of Hong Kong, Hong Kong 999077, China; gp.hancke@cityu.edu.hk; 3Division of Industrial Systems, RISE—Research Institutes of Sweden, 852 33 Sundsvall, Sweden; ulf.jennehag@ri.se

**Keywords:** IoT group key, quantum-safe cryptography, lightweight cryptography, lattices, lattice-based crypotgraphy, lattice trapdoors, one-way function, learning with errors, LWE, short basis

## Abstract

Group communication enables Internet of Things (IoT) devices to communicate in an efficient and fast manner. In most instances, a group message needs to be encrypted using a cryptographic key that only devices in the group know. In this paper, we address the problem of establishing such a key using a lattice-based one-way function, which can easily be inverted using a suitably designed lattice trapdoor. Using the notion of a bad/good basis, we present a new method of coupling multiple private keys into a single public key, which is then used for encrypting a group message. The protocol has the apparent advantage of having a conjectured resistance against potential quantum-computer-based attacks. All functions—key establishment, session key update, node addition, encryption, and decryption—are effected in constant time, using simple linear-algebra operations, making the protocol suitable for resource-constrained IoT networks. We show how a cryptographic session group key can be constructed on the fly by a user with legitimate credentials, making node-capture-type attacks impractical. The protocol also incorporates a mechanism for node addition and session-key generation in a forward- and backward-secrecy-preserving manner.

## 1. Introduction

As the Internet of Things (IoT) becomes an integral component of an ever-increasing number of application areas, the need to secure it is also assuming an ever-increasing level of importance. Security in the IoT is a multi-faceted problem that includes the usual security concerns regarding authentication, integrity, and confidentially; and IoT-specific concerns, such as how to design and implement cryptosystems on small devices; how to prevent, and recover from, attacks that leverage physically accessing an IoT device deployed at a physically accessible place; and preventing, and recovering from, denial-of-service attacks [1].

In this paper, we focus on the problem of how to establish a cryptographic group key that can be used to encrypt or decrypt a group message to ensure confidentiality in a dynamic IoT network. Group communication is an efficient and fast message-sending mechanism, whereby a message is broadcast to a group of users that constitute a given secure network. In contrast to unicast communication, where a sender sends a message to each potential recipient, in group communication, a single message is broadcast to potentially many recipients. This helps minimize the amount of traffic generated and computations performed at the sender’s end, since only a single message is broadcast to all potential recipients. In an IoT environment, where resources are generally constrained, minimizing traffic and computations at both the sender’s and the receiver’s ends is a desirable requirement. This model of communication would, for example, be desirable in a time-critical IIoT system where a controller must send a command to a group of actuators [2].

Below are some use-case scenarios were group communication would be desirable:*Smart lighting*: A smart building may have its lighting devices grouped according to their location and connected to a switch, which acts as a gateway. It is important that the switch is able so send group messages to the devices to control lighting levels and related functions;*Software updates*: A gateway downloads a software update and simply broadcasts it to the group so that member nodes patch. The alternative is each device downloads the patch independently, which results in generating unnecessary traffic;*Emergency broadcasts*: The control center of some automation may be forced to stop or start many devices in the process with a single command, minimizing time and resource requirements;*e-Health*: A sensor implanted in a patient’s body may broadcast readings to a group of receivers, such as nurses, doctors, and even chat servers.

Figure 1 shows an example where secure group communication would be preferable to unicast communication. Rather than send a message to each user in the secure multicast group, which is inefficient and computationally expensive, the *central entity* simply broadcasts an encrypted message that each user in the secure multicast group can decipher. The ability to send a single message to multiple recipients is also a desired property in time-sensitive applications, since sending a broadcast message is generally faster than sending a unicast message to each potential recipient.

A critical problem in enabling group communication is designing an efficient mechanism to ensure the confidentiality and integrity of group messages [4]. The standard mechanism to address this problem is to have all users in a group share a secret cryptographic key, so that only a user that has a copy of the secret key can decrypt a group message. This raises the problem of how to establish the secret key among the users in the first place, and how to initiate re-keying securely. There have been various cryptographic group-key establishment protocol proposals that rely on cryptographic constructions that do not have conjectured resistance against quantum attacks [5,6]. Recently, there has been a push towards basing cryptographic constructions on mathematical problems that are believed to be computationally hard, even for a quantum computer [7]. Constructing cryptographic primitives from such mathematical problems has the apparent advantage of both preparing for a future where quantum computers are a reality and having cryptographic systems whose security comes from alternative sources other than the usual ones, such as discrete logarithm [8]. Lattice-based cryptographic constructions in particular have gained an increased level of interest from the research community [9].

A lattice-based cryptographic construction has several advantages. On the security front, it provides an opportunity to build cryptosystems based on a worst-case security assumption, as opposed to constructions based on classical problems, such as integer factorization (IP), that are based on average-case hardness [10]. A cryptosystem based on worst-case hardness security has the interesting characteristic of being secure as long as a given computational problem is hard in the worst case, even for just one instance of the problem. A lattice-based constructions also has conjectured resistance to quantum-based attacks, as opposed to cryptosystems based on classical problems, which are easily breakable using quantum algorithms, as Shor [11] has demonstrated. In terms of richness, lattices enable the construction of cryptographic primitives and services that are otherwise impossible, such as homomorphic encryption. Regarding efficiency and ease of implementation, lattice-based computations involve simple linear-algebra computations, such as matrix-vector multiplication—which can be computed in parallel—and evaluations of simple linear functions.

Motivated by the aforementioned conjectured assumptions, in this paper, we present a cryptographic group-key establishment mechanism that is based on computational problems on lattices, for which there are no known efficient classical- or quantum-based algorithms. In particular, the cryptographic group-key establishment protocol is based on the concept of a lattice trapdoor, which helps us invert a lattice-based one-way function efficiently. Without the trapdoor, reverting the lattice-based one-way function is as hard as the search version of the well-known learning with errors (LWE) problem [12].

The main contributions of this paper are the following:A new method of designing a cryptographic group-key management protocol from lattice trapdoors is presented. Lattice trapdoors have been shown to be extremely versatile for designing various cryptographic primitives such as digital signatures and identity-based encryption (IBE) schemes [13]. The work presented here is a new addition to the list of cryptographic objects that can be built from lattice trapdoors. Since the computations involved are inherently lightweight, the protocol can be implemented and deployed in various IoT environments; hence, they contribute towards preparing the IoT for a future where quantum computers are a reality;A new mechanism for cryptographic group-key establishment where the group key is not stored in any of the constituent devices, so that an attacker cannot learn the cryptographic group key by physically examining a given device is also presented here. Moreover, we exhibit efficient mechanisms for adding or removing users from and to a secure group, in a manner that maintains standard security requirements, such as forward and backward secrecy.

The rest of this paper is organized as follows. In Section 2, we briefly discuss related work. In Section 3, we present basic lattice concepts that are essential for our subsequent discussion. In Section 4, we describe our network model for which the group-key establishment is proposed. In Section 5, we present the proposed protocol. In Section 6, we discuss the security and correctness of the proposed scheme. In Section 7, we discuss some implementation and optimization issues. In Section 8, we conclude the paper.

## 2. Related Work

The design and implementation of cryptographic group-key management protocols for the IoT has received a fair amount of attention from the research community [3,5,14]. The focus thus far has been on designing lightweight cryptographic group-key management protocols based on conventional cryptographic primitives, which are not quantum-safe. Transport layer-specific group-key establishment protocols, such as [14], suffer from the problem that they are not transport-layer-agnostic, in addition to not being quantum-safe. Other proposals, such as those based on elliptic curves [5], also lack the conjectured security against quantum-based attacks and do not scale well since they rely on a trusted anchor. Protocols such as [3,15] are lightweight and very convenient for the IoT, but are designed for one-to-many communication only, and are based on conventional constructions that are not quantum-safe.

Turning to lattice-based constructions, Lei et al. [16] have proposed an NTRU-based key-exchange protocol similar to the well-studied Diffie–Hellman key-exchange protocol [17]. An obvious limitation of this protocol is that it cannot be extended to a group consisting of more than two users. The identity-based key exchange from the lattices protocol proposed in [18] also cannot be used in settings where there are more than two users in a network. In [19], a quantum-cryptography-based key-management protocol has been proposed without considerations for the resource-limited nature of devices on which it is to be implemented. Likewise, the quantum key distribution proposed in [20] is not intended for resource-constrained devices. In [21], the authors present an NTRU-based key-generation algorithm, but do not consider the problem of generating session keys or adapting the protocol to resource-constrained devices. The work in [22] addresses a related problem of how to secure IoT-like social networks based on the blockchain.

To the best of our knowledge, there does not appear to be any cryptographic group-key-management protocols based on lattices or any other quantum-safe primitive designed for resource-constrained IoT devices in the literature.

## 3. Preliminaries

### 3.1. Notation

We use bold lower-case letters, such as x, to denote column vectors; for row vectors, we use the transpose $x^{t}$. Bold upper-case letters, such as A, denote matrices, while $A^{t}$ represents its transpose. Ax denotes the usual matrix-vector multiplication, and 〈a, b〉 represents the usual inner product of vectors a and b.

### 3.2. Lattice Trapdoors

We recall some basic definitions and theorems—without proofs—that are relevant to our discussion of the protocol. A more rigorous exposition of basic concepts about lattices can be found in [13] or [23], and the computational complexity analyses of various lattice-based computational problems can be found in [24].

**Definition** **1.** 
*A lattice L is the set of all integer linear combinations of some linearly independent basis vectors $B=\{b_{1},b_{2},\cdots ,b_{k}\}$. Formally,*

$$L=L\left(B\right):=BZ^{k}=\left\{\sum_{i=1}^{k}z_{i}b_{i}|z_{i}\in Z\right\},$$

*where B is the matrix whose columns are the n-dimensional column vectors $b_{1},b_{2},\cdots ,b_{k}$ [13].*


A given lattice L can be generated by infinitely many *bases*. Any two bases $B_{1}$ and $B_{2}$ of a lattice L are related by a *unimodular* integer matrix U such that $B_{1}=B_{2}U$, and vice versa with a different U [25].

A notion of *goodness* can be associated with lattices: a basis is considered *good* if its vectors are *short* under some reasonable notion of *norm* (typically the Euclidean norm), and are orthogonal or close to orthogonal to each other. A basis is otherwise considered *bad* [26]. As we will see later, some lattice problems, which are computationally hard on their own, become computationally easy if one has access to a good basis of the lattice. It is this important fact that we will mainly use in our construction.

**Definition** **2.** 
*The minimum distance of a lattice L is the length of a shortest nonzero lattice vector:*

$$\lambda_{1}\left(L\right):=\min_{v\in L\setminus \{0\}}∥v∥.$$



In general, $\lambda_{i}\left(L\right)$ is the smallest *r* such that L has *i* linearly independent norm vectors at most *r*. Here and everywhere else in this text, $∥.∥$ denotes the Euclidean norm.

**Theorem** **1** (Intersection of Lattices).
*Given two integer lattices $L(B_{1})$ and $L(B_{2})$, the set $L\left(B_{1}\right)\cap L\left(B_{2}\right)$ is also a lattice whose basis can be computed efficiently, i.e., in polynomial time in the dimension of the lattices. It is easy to prove by induction that this can be extended to any finite number of lattice bases [24]. Computing the intersection of two lattices is generally a slow process, but in our proof-of-concept implementation, we have devised a relatively fast process described as follows: to compute $L\left(B_{1}\right)\cap L\left(B_{2}\right)$, first compute the dual $L{\left(B_{1}\right)}^{*}$ and $L{\left(B_{2}\right)}^{*}$ of $L(B_{1})$ and $L(B_{2})$, respectively. Then, $HNF(B_{1}^{*}\cap B_{2}^{*})$ is the intersection of the two lattices, where HNF is the Hermite normal form, and $B_{1}^{*}$ and $B_{2}^{*}$ are the bases of $L{\left(B_{1}\right)}^{*}$ and $L{\left(B_{2}\right)}^{*}$, respectively.*


**Definition** **3** (Learning with errors (LWE)).
*Let X be a discrete Gaussian of a small width [27]. For a vector $s\in Z_{q}^{n}$, called the secret, the LWE distribution $A_{s,X}$ over $Z_{q}^{n}\times Z_{q}$ is sampled by choosing $a\in Z_{q}^{n}$ uniformly at random, choosing e←X, and outputting the vector (a, b=〈s, a〉+emodq) [12].*


### 3.3. Hard Lattice Problems

The following lattice problems—among many other lattice problems—are conjectured to be computationally hard, in the sense that there are no known classical or even quantum algorithms that can efficiently solve them.

**Definition** **4** (Search-LWE).
*Given m LWE samples, find the vector s.*


**Definition** **5** (Decision-LWE).
*Given m independent samples $\left(a_{i},b_{i}\right)\in Z_{q}^{n}\times Z_{q}$, where every sample is distributed according to either $A_{s,X}$ (for a fixed s) or the uniform distribution, distinguish which one of the two is the case.*


**Definition** **6** (Bounded distance decoding problem (BDD_γ_)).
*Given a basis B of an n-dimensional lattice L=L(B), and a target $t\in R^{n}$, with the guarantee that $dist\left(t,L\right)<\lambda_{1}\left(L\right)/\left(2\gamma \left(n\right)\right)$, find the unique vector v such that $∥v-t∥<d$, where γ is some function of n, where n is the dimension of L.*


One can easily show that search-LWE can be cast as an *average-case* BDD problem on the following family of lattices:(1)$$L\left(A\right):=\left\{A^{t}s:s\in Z_{q}^{n}\right\}+qZ^{m},$$
where A is the matrix whose columns are the $a_{i}\in Z_{q}^{n}$ samples.

Although the BDD problem is computationally hard on its own, with a proper *trapdoor*, it can be easily solved. We refer the reader to [24] for a rigorous analysis of the computational complexity of each problem, and various classical and quantum reductions between different problems.

## 4. System Model and Security Requirements

The system consists of a set of IoT users, denoted by a universe U, as depicted in Figure 2. Each user in the system is denoted by $U_{i}$, with *i* ranging from 1 to *n*, where n=|U|. A secure multicast group is a subset G of U. The aim of the proposed protocol is to enable each user in G to agree on a shared secret. Note that G is dynamic, since we want to add or remove users to and from the group.

We assume that each pair $U_{i}$ and $U_{j}$, i≠j, in U have an authenticated channel between them. This assumption allows us to rule out an active adversary that attempts to break the protocol by injecting messages by masquerading as a legitimate user during the key-establishment process. Although this assumption abstracts the practical problem of authentication away, it allows us to focus only on aspects related to the secure cryptographic group-key-establishment mechanism.

To simplify the analysis of the security of the system, we assume a reliable communication infrastructure (e.g., no packet loss due to interference).

Regarding security requirements, the first requirement is that a passive adversary—i.e., an adversary that attempts to learn something useful by only observing what is out in the open [28]—cannot learn the shared secret key without solving a computational lattice problem that is conjectured to be hard.

The second requirement, which follows from the first as a corollary, is that a passive adversary cannot decrypt a group message that was encrypted using the shared secret key without solving a computationally hard lattice problem. We also require that a new user is added to or removed from the group in a manner that preserves forward and backward secrecy [29]. Finally, we require that group sessions keys be independent from each other to achieve forward and backward secrecy.

In summary, the security requirements are the following:**Requirement 1**: Any user $U_{i}\notin G$ cannot learn the group key without solving a hard lattice problem;**Requirement 2**: Each session group key is independent from any other session key;**Requirement 3**: A user $U_{i}$ is added to G in a forward- and backward-secrecy-preserving manner.

The methodology consists in constructing a scheme and proving that it satisfies the above-mentioned requirements, assuming that some lattice-related problems are computationally hard. To demonstrate feasibility, a toy version of the protocol will be implemented and results analyzed.

## 5. Proposed Scheme

We start our presentation of the protocol by introducing the concept of a lattice trapdoor.

### 5.1. Setting Up a Lattice Trapdoor

Informally, a lattice trapdoor is a piece of information that enables one to invert a one-way function defined on a lattice, which is otherwise hard to invert on its own. A lattice trapdoor, in particular, enables one to solve the BDD problem.

Generally, there are two types of lattice trapdoors: *short bases* and *gadget trapdoors* [30]. We describe the proposed protocol using the notion of a short basis as trapdoor, although one can also use gadget trapdoors without having to change much in the protocol construction.

An important property of any given lattice is that it can have arbitrarily many (short) bases. Any one of the short bases can be used to invert a one-way function defined on that particular basis. If one can devise a mechanism for generating a lattice along with many short bases—one for each user—and securely save each short basis in each user, then each user can use its short basis to invert the one-way function and extract a secret value, which can then be used as a group key. That is the main idea behind the proposed scheme.

The first part of the protocol deals with setting up a trapdoor for each user $U_{i}$ in the group G. Although it is likely that each user will end up having a different trapdoor, we will show that each trapdoor will enable each user to invert a common one-way function. This is at the heart of the protocol—the fact that different but related trapdoors can be used to invert a given one-way function.

The lattice-based one-way function that we will rely on is the following LWE function:(2)$$g_{A}\left(s,e\right):=s^{t}A+e^{t}\;\mathrm{mod}\;q,$$
where $s\in Z_{q}^{n}$ is chosen uniformly at random, and $e\in Z^{m}$ is a small vector chosen from the LWE error distribution $X^{m}$. Inverting this function is equivalent to the search-LWE problem [12]. With a trapdoor information, the function can easily be inverted. The trapdoor is a short basis S of the lattice
$$L\left(A\right):=\left\{A^{t}s:s\in Z_{q}^{n}\right\}+qZ^{m}.$$

With a short basis S, the function in Equation (Equation 2) can easily be inverted (i.e., s and e can be recovered), given that the the parameters are chosen appropriately [30].

There are two challenges associated with setting up a trapdoor corresponding to the function in Equation (Equation 2). The first challenge is how to generate a random lattice L(A) and its corresponding short basis S. The second challenge is how to generate different short bases $S_{i}$ for each $U_{i}\in G$, each corresponding to the lattice L(A).

In order to generate a random lattice along with a short basis, we use Ajtai’s method [31]. We recall it here as a theorem:

**Theorem** **2.** 
*There is an efficient randomized algorithm that, given positive integers n,q and m≥Cnlogq for some constant C, outputs a nearly uniformly random matrix $A\in Z^{n\times m}$ specifying the integer lattice $L=L^{⊥}\left(A\right)=\left\{x\in Z^{m}:Ax=0\;\mathrm{mod}\;q\right\}\subseteq Z^{m}$, along with a basis $S\in Z^{m\times m}$ of L whose vectors have norms bounded by poly (n,logq).*


It is worth noting that the lattice in Theorem 2 corresponds to the short integer solution (SIS) problem [32], and is a dual [33] of the lattice in Equation (Equation 1). Ajtai’s algorithm is slow and complex for small IoT users. However, Micciancio et al. [30] have improved upon Ajtai’s algorithm. Due to [30], it is now possible to generate a random SIS lattice along with a corresponding short basis S efficiently and easily. This addresses the first challenge.

Regarding the second problem, assume that the users in G are indexed $U_{1}$ through $U_{k}$, where k=|G|. Without loss of generality, we may assume that $U_{1}$ is the group leader, by which we mean that it starts the trapdoor-setup process.

User $U_{1}$ starts the trapdoor-setup process by broadcasting a *group-join* message. The *group-join* message consists of a group ID chosen by user $U_{1}$ and a short description of the group. The *group-join* broadcast message serves as a signal to to each user $U_{i}\in U$ to join the group. We assume that a subset G of the users in U will respond to the *group-join* message positively, i.e., will want to join the group.

Each user that wants to join the group simply broadcasts its ID. After a predefined amount of time has elapsed, the leader broadcasts the set $G=\{ID_{1},ID_{2},\cdots ID_{k}\}$ of all received IDs. The set G gives each user $U_{i}$ information about the user whose ID precedes it. Broadcasting the set G helps the users in G form a logical ring structure in which each user knows the user before it, assuming that a notion of ordering can be associated with respect to the IDs.

User $U_{1}$ starts the trapdoor-setup process by generating a lattice $L_{1}=L_{1}^{⊥}\left(A\right)=\left\{x\in Z^{m}:Ax=0\;\mathrm{mod}\;q\right\}\subseteq Z^{m}$, along with a short basis $S_{1}$, using the mechanism stated in Theorem 2. It keeps $S_{1}$ secret and makes $L_{1}^{⊥}\left(A\right)$ public. Note that this corresponds to the idea of generating a *good* and *bad* basis pair, so that the *good* basis is used as a trapdoor to a one-way function based on the *bad* basis.

Each user $U_{i}$, where *i* ranges from 2 to *k*, does the following: upon receiving $L_{i-1}$, it generates the $L_{i}$ and $S_{i}$ pair as before, but with the additional requirement that $L_{i-1}\cap L_{i}\neq \emptyset$. The problem of how to generate such lattice pairs has been addressed in [25]. We use a slightly modified version of the algorithm discussed in [25] (please see Algorithm 1). Given a lattice with a bad basis, Algorithm 1 generates a lattice with a Hadamard ratio [34] of 0.75, which is considered a good basis. Figure 3 depicts a lattice with a good basis generated from a lattice with a bad basis (a Hadamard ratio of 0.37). In our proof-of-concept implementation, we have used a less-optimized version of one of the algorithms proposed in [25]. User $U_{i}$ keeps $S_{i}$ private and makes $L_{i}\cap L_{i-1}$ public. This round finishes with $U_{k}$ generating $S_{k}$ and its corresponding $L_{k}$, with the requirement stated above. At the final round, $U_{k}$ makes $L_{k}\cap L_{k-1}$ public. We call this last lattice $L_{G}$. The trapdoor setup process is shown in Algorithm 2.

**Algorithm 1:** A Short-Basis Lattice from a Long-Basis Lattice 
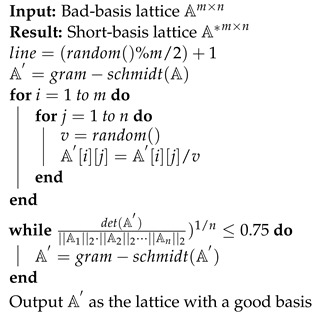


**Algorithm 2:** Trapdoor Setup 
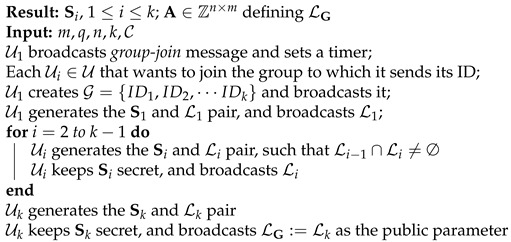


We remark that $L_{G}$ is represented by the matrix $A\in Z^{n\times m}$, which defines a *bad* basis for the lattice
$$L\left(A\right):=\left\{A^{t}s:s\in Z_{q}^{n}\right\}+qZ^{m}.$$

The matrix A is a public parameter, and can be suitably stored in a public directory which each user $U_{i}$ can access, or be stored locally in each user.

See Figure 4 for an instance of the trapdoor-initialization process with four users.

### 5.2. Group Key from a One-Way Function with a Trapdoor

First, we note that $L_{G}\subseteq L\left(S_{i}\right),1\leq i\leq k$. This is true because $L_{G}=L\left(S_{1}\right)\cap L\left(S_{2}\right)\cap \cdots \cap L\left(S_{m}\right)$, which is a subset of $L(S_{i})$, for each i∈{1,2,⋯,k}.

By construction, $L_{G}$ is a lattice of the form
$$L_{G}=\left\{x:\mathrm{Ax}=0\right\}$$
with parameters as specified in Theorem 2, and is specified by the parity-check [35] integer matrix A.

Our one-way function is the following
(3)$$g_{A}\left(s,\;e\right)=s^{t}A+e^{t}\;\mathrm{mod}\;q,$$
where the matrix A is constructed as defined in Section 5.2, and e, s and q are chosen according to the distributions specified in Definition 5.

### 5.3. The Group Key and Group-Message Encryption

The group key is some function *f* of $s\in Z_{q}^{n}$. The function *f* could be a pseudo-random function that takes s as its seed and expands it to a required bit-length suitable for the encryption algorithm agreed upon in the group. It could well be a function that truncates certain bits from s.

More formally, let E be the symmetric-key-encryption algorithm used to encrypt group messages, and let M be the message space. To encrypt a message m∈M, $U_{i}$ chooses $s\in Z_{q}^{n}$ and an error $e\in Z^{m}$, according to the distributions specified in Equation (Equation 1), and outputs
(4)c=E(f(s),m);

$U_{i}$ then computes the one-way function $g_{\left(s,e\right)}$ and appends it to *c*. The final combined message is then simply broadcasted. Algorithm 3 describes how the encryption process works.

In the next section, we prove that only a user in G can recover s and, hence, decrypt the encrypted group message *c*.
**Algorithm 3:** Group-Message Encryption**Result**: Encrypted group message c,n,q**Input**: Encryption algorithm E, message *m*, pseudo-random function *f*Choose $s\in Z_{q}^{n}$ randomlyCompute key=f(s)Compute c=E(key,m)Broadcast *c* as the encrypted group message

### 5.4. Decrypting a Group Message

Decrypting a group message *c* involves first recovering *s* and then using the agreed-upon decryption algorithm to recover *m*.

To recover *s*, each user $U_{i}\in G$ uses its short-basis trapdoor $S_{i}$ to compute:$$\begin{array}{c} x^{t}=b^{t}\left(S_{i}\right)=e^{t}\left(S_{i}\right)\;\mathrm{mod}\;q, \end{array}$$
where
$$\begin{array}{c} b^{t}=g_{A}\left(s,\;e\right)=s^{t}\left(A\right)+e^{t}\;\mathrm{mod}\;q \end{array}$$

The vector x is then lifted to its canonical representative [26] $\bar{x}\in {\left[-\frac{q}{2},\frac{q}{2}\right)}^{m}$, from which we obtain
$$\begin{array}{c} e={\bar{x}}^{t}\left(S^{-1}\right) \end{array}$$

Computing s, given e, is straightforward, as shown in [13].

### 5.5. Adding a New Node to a Group

In a dynamic IoT network, there could arise a need to add a new user $U_{i}\in U\setminus G$ to G. A standard security requirement in such a scenario is that $U_{i}$ does not learn any of the s values used before it joins the group, nor is any group message exchanged prior to joining. This is the usual backward-secrecy requirement [36].

Node addition is effected as follows: $U_{i}$ reads off the public parameter $L_{G}$. It then generates an $S_{i}$ and $L_{i}$ pair, with the requirement that $L_{i}\cap L_{G}\neq \emptyset$. Finally, node $U_{i}$ broadcasts $L_{i}\cap L_{G}$ as the new $L_{G}$. All other nodes in G update their copy of $L_{G}$ accordingly. The node-addition process is shown in Algorithm 4.

Notice that each user $U_{j}\in G\setminus \left\{U_{i}\right\}$ does not update its copy of $S_{j}$ when $U_{i}$ is added to G.
**Algorithm 4:** Adding a Node $U_{i}$ to G 
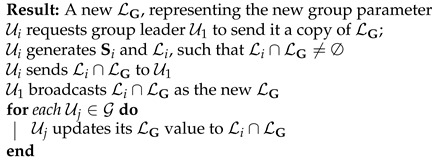


### 5.6. Removing a Node from a Group

Once a secure group has been established, the need to remove a node $U_{i}\in G$ could emerge for various reasons [37]. A common security requirement in such a scenario is that $U_{i}$ be unable to read future messages or have access to any of the future session keys [38].

In the proposed protocol, the only way to remove a node while maintaining the said security requirement is to re-run the protocol from scratch, excluding the node to be removed. This is a straightforward but inefficient method. An improved solution is left for future work.

### 5.7. Generating Session Keys

Generating session keys in such a way that security requirements such as forward and backward secrecy are maintained is not an easy problem. The proposed protocol easily solves these problems since the session key is a value derived from s, where s is chosen uniformly at random at each round. That is, if we let *t* represent the round number, the $s\in R^{n}$ used at round $t_{i}$ is independent from the $s\in R^{n}$ used at round $t_{i-1}$ or $t_{i+1}$. So, if an attacker ever succeeds in learning the value of s at any particular round, the attacker does not learn anything about previous or future values of s.

In each round *t*, a user $U_{i}$ that intends to broadcast a group message *m* chooses $s_{t}\in Z_{q}^{n}$ randomly, computes $c=E(f\left(s_{t}\right),m)$, and broadcasts *c*. A user $U_{j}$ uses its short basis $S_{j}$ to recover $s_{t}$ and runs the decryption algorithm using $f(s_{t})$ as the decryption key. If *t* and t+1 represent two consecutive rounds, or group sessions, then knowledge of $s_{t}$ does not help an attacker learn anything about $s_{t+1}$, nor does knowledge of $s_{t+1}$ reveal anything useful about $s_{t}$. If each $s_{i}$ is chosen sufficiently randomly and the LWE function is indeed hard, then independence of session keys from each other follows easily.

## 6. Proof of Correctness and Security

By *correctness*, we mean that each $U_{i}$ in G ends up recovering the same value of s. By *secure*, we mean that no user U∉G can either classically or quantumly recover s. We state each claim as a theorem and provide a proof for each.

**Theorem** **3** (Correctness).
*The group-key-agreement protocol is correct.*


**Proof.** This amounts to proving that each $S_{1}$ is a good (that is, short) basis for $L_{G}$. Strictly speaking, this is generally not the case since $L\left(S_{i}\right)⊃L_{G}$, for each i∈{1,2,⋯,m}. However, since $L_{G}\subseteq L\left(S_{i}\right)$, and since each $S_{i}$ is a short basis by construction, each $S_{i}$ is a trapdoor for $g_{\left(s,e\right)}$. The details of how s can be recovered given a short basis S can be seen in [13] or [39]. □

**Theorem** **4** (Security).
*The group-key-agreement protocol meets the security requirements specified in Section 4.*


We only consider a passive adversary; that is, we consider an adversary that can attempt to learn the group key or some partial information about the group key by intercepting data that is in the open. We do not consider an active adversary.

**Requirement 1**: An attacker that attempts to recover one of the short bases $S_{i}$ or generate a new short basis for the lattice $L_{G}$ has to solve a version of the (approximate) shortest vector problem (SVP) [40], which we conjecture to be computationally hard. An attacker that attempts to directly recover the secret s by inverting the one-way function *g* has to solve a version of the BDD problem, since the two problems are computationally equivalent. The BDD problem is again conjectured to be computationally hard. Therefore, a passive adversary cannot recover s without solving either one of the two lattice problems. Since we conjecture the problems to be hard even for quantum computers, we conclude that the protocol is quantum-safe. The confidentiality of a message *m* encrypted using *s* follows as a corollary, assuming that the encryption scheme E is quantum-safe. The encryption scheme could be a standard protocol, such as the AES, with a proper key length to account for possible quantum-based attacks;**Requirement 2**: This easily follows from the fact that, at each round, a user $U_{i}$ chooses s randomly;**Requirement 3**: By design, the addition of a new node $U_{j}$ into G forces each node $U_{i}\in G$ to acquire a new $L_{G}$ that is not related to the previous $L_{G}$ or a future $L_{G}$.

## 7. Results and Performance Analysis

We have implemented a proof of concept of the protocol on a simulated IoT network on Contiki OS [41]. Using a 300×300 full-rank lattice, which would be insecure for practical purposes due to the small parameter sizes, each node takes 4.94 s on average to set up a trapdoor. The restriction to small lattices was motivated by the memory-size-related limits of the IoT-device simulators provided by Contiki [22]. One can extrapolate the encryption and decryption times for larger lattices for the results obtained for smaller ones.

Encryption takes 0.023 s on average and decryption takes 3.04 s on average. With proper optimization, these numbers can be improved, making the protocol fast.

Figure 5 depicts the running times for setting up a trapdoor on each node, encrypting, and decrypting (128-bit key, 128-bit message) an IoT network consisting of 10 Raspberry Pi 3Bs [42].

Regarding error rate, each node recovers the same s value—on average—92 percent of the time, meaning that 8 times out of 100, decryption does not work. This is not a significant problem, however, since one can use a reconciliation technique [43] to resolve any discrepancies in decryption.

The main positive aspect about the performance of the protocol is that once a trapdoor is set up on each node, all the other functions—encryption, decryption, session-key generation, and node addition—are effected in constant time, regardless of the size of the network. Setting up trapdoors takes Θ(n), where *n* is the number of nodes in G. This is so because the setting up of trapdoors is conducted in a sequential manner, but in principle, it can be effected in parallel. The fact that most of the main functions of the protocol can be achieved in constant time is a considerable improvement over other protocols available in the literature, which take at least linear time in the size of the input [6]. Operations that take constant time are especially desirable in larger networks.

In concrete terms, setting up a trapdoor, the most time-consuming function, took less that 5 s, albeit in a network with 10 nodes and a small lattice (300 by 300 dimensional lattice). The other key-agreement-related functions took less than a second to complete.

Analyzing the protocol in relation to other protocols that solve the same or a related problem is impractical due to the specificity of the design—different primitives, different setups, and different construction. At an abstract level, our protocol outperforms related protocols with respect to the running time and amount of storage required. See Table 1 for the running times of each function of the proposed protocol. All the protocols discussed in Section 2 that deal with the same problem have at least a Θ(n) running time for every function. This is, of course, a crude comparison, because the operations involved in each function are different. Regardless, a constant running time is a considerable improvement, especially in large networks. With respect to storage, our protocol does not require a user to store anything permanently since a session is constructed on the fly.

In this work, we only presented the basic building blocks of the protocol, without regard to how it can actually be implemented efficiently. One issue with lattice-based cryptographic constructions in general is the size of the security parameters. For a reasonable security level, such as a 128-bit security level, the dimension of the lattice should be greater than 500 [26]. This means that each device would need to store at least 2 matrices, each in the order of 100s of kilobytes. This could be an issue for extremely small devices with highly constrained memory sizes. This problem can be partially solved by using the Hermite normal form (HNF) of the matrix A that describes the lattice $L_{G}$ [25]. This optimization reduces the size of the matrix by a factor of *n*, where *n* is the dimension of the lattice. A further improvement both in terms of storage and computation can be achieved by defining the one-way function on the ring-LWE problem, which is more compact and allows for faster computations using fast Fourier transform (FFT) techniques [44]. One can then use Micciancio’s compact one-way function, which is syntactically similar to the one presented here [45].

## 8. Conclusions and Direction for Future Work

In this paper, we have presented a new cryptographic group-key-management protocol based on lattice trapdoors. The protocol enables a group of IoT devices to define a trapdoor-based one-way function that only a device in the network can revert. We have shown how one can construct a cryptographic group-key from the one-way function.

The results show that the protocol can be implemented on resource-constrained IoT devices. Moreover, the fact that most of the functions of the protocol can be effected in constant time is an important improvement on similar protocols that attempt to solve the same problem.

The advantage of the protocol is twofold: First, it is highly lightweight since the computations involved are simple linear-algebra operations that are parallelizable and evaluations of simple linear functions. Second, it provides conjectured security against potential quantum-based attacks and has security based on worst-case hardness assumptions. We have also demonstrated how to add a user to a secure group dynamically, while maintaining the necessary security requirements, all in constant time, regardless of the size of the network. The protocol can be potentially deployed on various IoT networks, and help secure such networks against future quantum-based attacks.

The protocol can be improved in future work by expanding the security model to include an active adversary, devising a parallel bad/good lattice basis-generation mechanism, and optimizing the implementation.

## Figures and Tables

**Figure 1 sensors-22-04148-f001:**
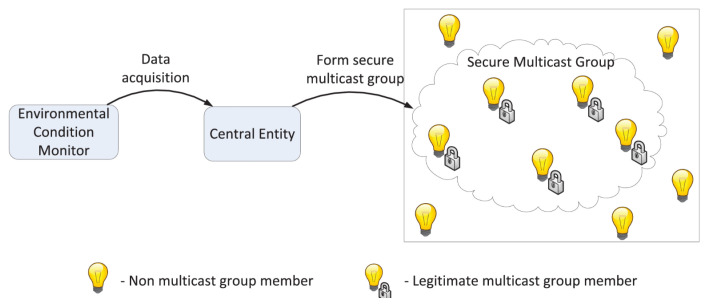
Multicast for light bulbs—Figure 1 of [3].

**Figure 2 sensors-22-04148-f002:**
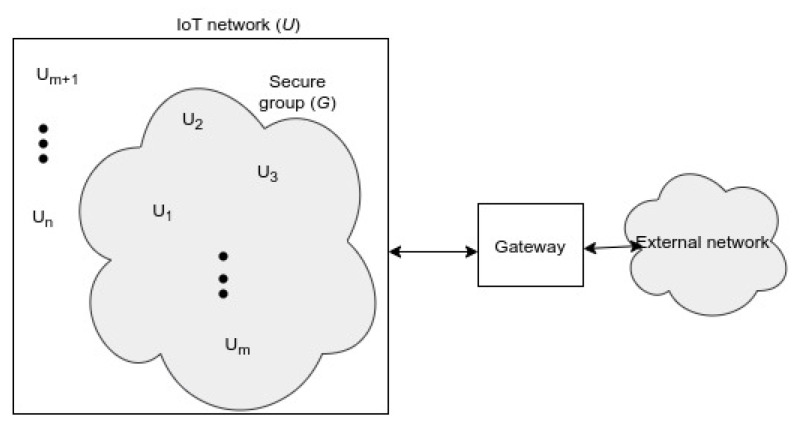
Network model. The network consists of a gateway and a set of nodes, supported by a communication infrastructure. All or a part of the nodes may be members of the secure group at a given time, as shown in the figure (*m* of *n* nodes are in the secure group).

**Figure 3 sensors-22-04148-f003:**
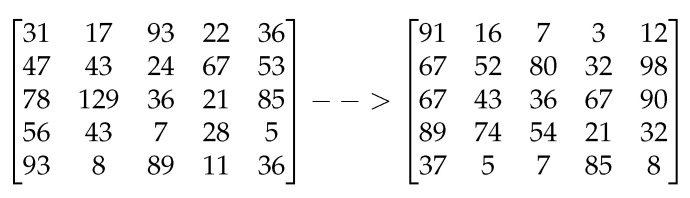
Public lattice → new private lattice.

**Figure 4 sensors-22-04148-f004:**
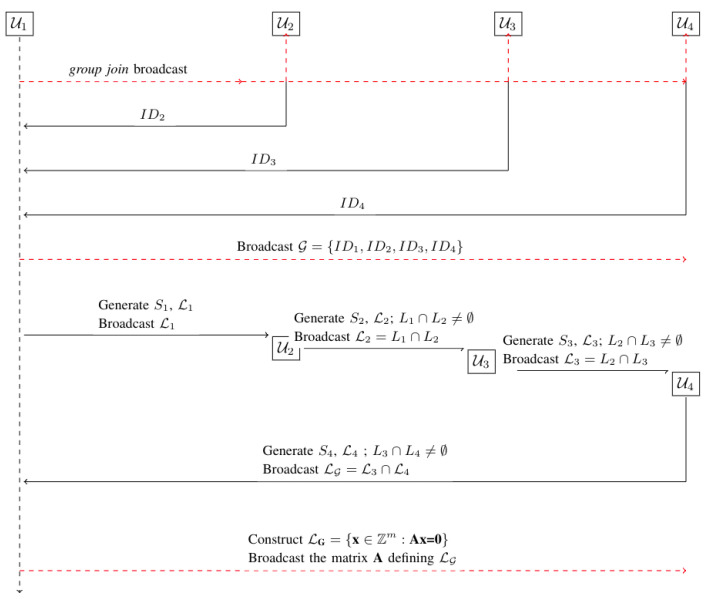
Trapdoor-setup procedure with four users.

**Figure 5 sensors-22-04148-f005:**
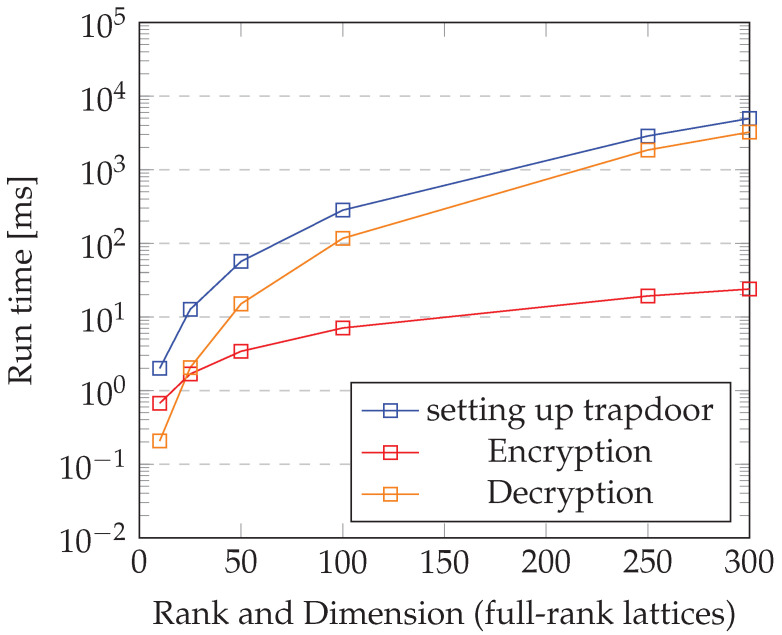
Running time.

**Table 1 sensors-22-04148-t001:** The table shows the running time for each function as a function of *n*, where *n* is the number of users in the secure group.

Function	Running Time
Setting up trapdoor	Θ(n)
Encrypting a group message	Θ(1)
Decrypting a group message	Θ(1)
Generating a session key	Θ(1)
Adding a node	Θ(1)
Removing a node	Θ(n)

## Data Availability

Not applicable.

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
