# Peer review of "Quantum-Safe Group Key Establishment Protocol from Lattice Trapdoors"

_sensors, 2022, doi:10.3390/s22114148_

Round 1
Reviewer 1 Report
The manuscript addresses one of the security issues used in IoT devices. Also, it considers a technique using lattice trapdoors. I have read the article, and I recommend it with small revisions. I recommend the authors add more details in Section 7, where the analysis shows some light details. Otherwise, I have no issue with this paper.
Reviewer 2 Report
The Authors propose a communication protocol designed for the Internet of Things (IoT). The proposed solution can be applied in resource-constrained IoT network systems. The necessary computations performed within the scheme frame involve simple parallelizable linear-algebra operations and evaluations of simple linear functions. What is relevant, the protocol exhibits resistance against potential quantum computer-based attacks. Additionally, the Authors convince that the scheme incorporates mechanisms allowing for the simple addition of the nodes and session-key generation. The Authors also present obtained parameters of an exemplary implementation of their proposal.
The manuscript is well written. It is clear and provides comprehensive information to the Readers, which helps them understand the ideas presented in the article and could be a basis for further research. The proposed scheme seems to be new, promising, and valid enough to be published. The proposed protocol is not only relevant from the theoretical point of view but also carries great application potential. Thus, one can conclude that the manuscript deserves publication and can be accepted in its present form.
Author Response
We have that reviewer for his helpful comments.